# Porous Structure and Fractal Dimensions of Activated Carbon Prepared from Waste Coffee Grounds

**DOI:** 10.3390/ma16186127

**Published:** 2023-09-08

**Authors:** Sofiia Victoriia Sklepova, Nataliia Ivanichok, Pavlo Kolkovskyi, Volodymyr Kotsyubynsky, Volodymyra Boychuk, Bogdan Rachiy, Andrzej Uhryński, Michał Bembenek, Liubomyr Ropyak

**Affiliations:** 1Department of Material Science, Vasyl Stefanyk Precarpathian National University, 57 Shevchenko Str., 76018 Ivano-Frankivsk, Ukraine; sonja93sklepova@gmail.com (S.V.S.); natalia.ivanichok@gmail.com (N.I.); volodymyr.kotsuybynsky@pnu.edu.ua (V.K.); bogdan_rachiy@ukr.net (B.R.); 2Department of Solid State Chemistry, V. I. Vernadsky Institute of General and Inorganic Chemistry, National Academy of Sciences of Ukraine, 32/34 Academician Palladin Ave., 03142 Kyiv, Ukraine; pavlo.kolkovskyi@pnu.edu.ua; 3Department of Physics, Vasyl Stefanyk Precarpathian National University, 57 Shevchenko Str., 76018 Ivano-Frankivsk, Ukraine; volodymyra.boichuk@pnu.edu.ua; 4Department of Machine Design and Maintenance, Faculty of Mechanical Engineering and Robotics, AGH University of Science and Technology, 30 Mickiewicza Ave., 30-059 Krakow, Poland; uhrynski@agh.edu.pl; 5Department of Manufacturing Systems, Faculty of Mechanical Engineering and Robotics, AGH University of Science and Technology, 30 Mickiewicza Ave., 30-059 Krakow, Poland; 6Department of Computerized Mechanical Engineering, Ivano-Frankivsk National Technical University of Oil and Gas, 15 Karpatska Str., 76019 Ivano-Frankivsk, Ukraine

**Keywords:** disordered carbon, micropore, mesopore, nitrogen porosimetry, fractal dimension, coffee

## Abstract

The present work reports the results of a systematic study on the evolution of the morphological properties of porous carbons derived from coffee waste using a one-pot potassium-hydroxide-assisted process at temperatures in the range of 400–900 °C. Raw materials and obtained carbons were studied by TG, DTG, SEM and nitrogen adsorption porosimetry. The decomposition temperature ranges for hemicellulose, cellulose and lignin as the main component of the feedstock have been established. It is shown that the proposed method for the thermochemical treatment of coffee waste makes it possible to obtain activated carbon with a controllable pore size distribution and a high specific surface area (up to 1050 m^2^/g). A comparative study of the evolution of the distribution of pore size, pore area and pore volume has been carried out based on the BJH and NL-DFT (slit-like pores approximation) methods. The fractal dimension of the obtained carbons has been calculated by Frenkel–Halsey–Hill method for single-layer and multilayer adsorptions.

## 1. Introduction

Today, the uninterrupted availability of a sufficient number of energy sources at an affordable price is a standard of living and safety for the population of the world. At the same time, the structure of the modern energy system must provide the ability to withstand all kinds of shocks, including natural disasters, global climate change, geopolitical and military conflicts, etc. These needs have led to the widespread use of coal, oil and natural gas, which creates several problems: depletion of fossil fuel reserves; excessive emissions of greenhouse gases; and a harmful ecological footprint of production [1,2,3]. Despite the listed problems, it is expected that in the coming decades, oil and gas will remain important components of world energy demand, especially in the transport sector [4]. However, energy systems structured in this way are a large source of anthropogenic greenhouse gases, so the decarbonization of the energy industry is becoming a key element of global policy on climate change [5,6].

Of all existing sources of energy, electricity occupies a leading position in meeting the needs of households, industry and ordinary consumers. To effectively solve the problems of excessive energy demands, reliable ways of storing it are required. The storage of electricity requires a material that has an acceptable cost and that can be synthesized using natural resources and available renewable energy sources [7,8,9,10].

Further progress concerning enhancing the performance of electric double-layer electrochemical capacitors (ultracapacitors) will be possible only after the development of new methods for obtaining porous carbon with optimized morphological properties [11]. The study of the evolution of a porous structure is crucially important for revealing the main effects of thermal and chemical treatments on the specific surface area and pore characteristics of activated carbons. An additional important factor is the type of raw material used to obtain carbons, such as fruit stones, coconut shells or flax fiber [12]. The chemical composition (the ratio of the content of lignin, cellulose and hemicellulose) as well as the structural order of plant feedstock determine the morphology of materials after carbonization and are a precondition for further changes during chemical activation [13]. The choice of a strategy for the preparation of activated carbon based on certain plant materials will make it possible to control its main morphological characteristics (pore size distribution, pore type, ratio of micro- to meso-pores and specific surface area) [14,15,16]. Increasing the performance of an ultracapacitor electrode requires the formation of both “transport” pores, the diameter of which exceeds the size of the solvated electrolyte ions, and “storage” pores, with diameters close to those of electrolyte ions. The search for an optimal balance between different types of pores makes it possible to optimize ion transport and form an electric double layer. Another important characteristic of activated carbon as a system consisting of turbostratically arranged carbon fragments is its fractal dimensions [17,18]. This parameter characterizes the degree of geometric complexity and structural heterogeneity of activated carbon as a rigid framework consisting of both micro- and meso-pores [19]. There are three different types of fractal analysis based on the analysis of the perimeter, surface and mass of fractal dimensions, respectively [20]. Fractal analysis is an effective method for describing the porous structure of porous carbons. Specialized optical interference filters have recently been successfully used to improve the accuracy of fractal analysis procedures [21,22,23]. The use of high-precision thin-film sensors based on single-crystal materials has improved the targeting of fractal analysis [24,25].

Moreover, the fractal dimension can be used to characterize the size and distribution of pores in the material, determine the geometric features of the pores and also compare various porous carbon materials in terms of their structural properties. The use of fractal dimensions makes it possible to obtain quantitative characteristics of the porous structure of carbon materials, which can be used for further analysis and modeling of their behavior. Carbon materials, including graphene, fullerenes, single-walled and multi-walled nanotubes and carbon aerogels, have unique fractal structures. The fractal structure of carbon materials is one of the key aspects that determine the mechanical, electrical and thermal characteristics of these materials. The study of the fractal structure of nanoporous carbon materials is of great importance for understanding their physical and chemical properties, as well as for the development or optimization of the methods of synthesis of these materials with controlled or even assigned characteristics, such as conductivity, band gap, catalytic activity and others [26,27,28]. The development of porous materials with a controlled morphology allows for expanding the possibilities of obtaining composite materials using the infiltration technique [29] with low melting point alloys [30], leading to desirable combinations of mechanical and tribological properties [31].

There are already a few works dealing with the synthesis of carbon from coffee grounds, for example [32], but a feature of our work is the approach of predicted optimization of the morphological properties of coffee-ground-biowaste-derived carbons. Compared to [33], the novelty of this work is finding general regularities of the fractal structure changes of biocarbons synthesized using a fast one-pot method. The typically used activated agents for preparation of biochar-derived activated carbon are H_3_PO_4_ and ZnCl_2_ as well as alkali metal carbonates and oxides [33]. The process of the chemical activation of biochar with hydroxides of alkali metals is complicated because the structural and textural parameters of activated carbons significantly depend on the activation procedure details [34]. Simultaneously, the use of potassium hydroxide as an activated agent with adjusting the activation procedure conditions (concentration, temperature and duration of the process, pretreatment carbonization) allows us to produce porous carbons with extremally high specific surface areas (up to about 3500 m^2^·g^−1^) combined with a high content of meso- and macro-pores [32]. As a result, the study of potassium-hydroxide-assisted thermochemical activation of coffee ground waste without precarbonization or additional chemical treatment is of significant scientific and practical interest.

This paper presents results on the effect of thermochemical activation conditions on the morphological characteristics and textural properties of activated carbon obtained from waste coffee grounds. Based on the analysis of nitrogen adsorption isotherms using the slit-like pore model, changes in the fractal dimension of microporous activated carbons are presented. Processing coffee ground waste into a porous carbon material allow us to effectively repurpose this waste and reduce its environmental impact according to the principles of a circular economy and waste reduction, contributing to a more sustainable approach to waste management. The ability to repurpose a common waste product into a valuable and versatile material underscores the importance of this approach for sustainable development and resource optimization. This approach offers versatile solutions across various sectors, including energy, environment and healthcare industries [35]. As an example, porous carbon derived from coffee grounds is a perspective electrode material in energy storage devices such as supercapacitors and batteries that can drive advancements in renewable energy technologies and electric vehicles, ultimately reducing the reliance on fossil fuels.

The purpose of this research is to develop a technology for the production of activated carbon from coffee waste and to study the effect of the synthesis conditions on its properties.

## 2. Materials and Methods

The production of activated carbon from waste coffee grounds was carried out according to the developed technological process (Figure 1).

Fresh coffee ground waste without fermentation was used as a raw material for the preparation of activated carbon without additional milling after washing and drying at a temperature of 90 °C for 48 h (Figure 2).

A mixture of dried coffee waste, KOH and water (mass ratio 1:0.5:1) was transferred to a stainless steel autoclave and sealed. To design the autoclave, we used the methodical approach proposed in paper [36]. Special attention was paid to the tightness and reliability of the threaded connections [37,38,39,40]. The mixture was heated in an autoclave at a rate of 10 °C/min. After reaching the thermochemical treatment temperatures (400, 500, 600, 700, 800 and 900 °C), the autoclave was kept under these conditions for 0.5 h. After isothermal exposure, the furnace was turned off and cooled to room temperature. The resulting carbons were repeatedly washed with distilled water and 5% aqueous HCl solution to pH = 5.0–5.5, dried at 90 °C for 48 h, grinded to a fraction size of about 100–150 μm (Figure 3) and labelled according to the thermochemical treatment temperature (°C) as SX, where X = 400, 500, 600, 700, 800 or 900.

Thermal analysis was performed on an STA 449 F3 Jupiter simultaneous thermal analyzer using NETZSCH Proteus^®^ 8.0 software (Erich NETZSCH GmbH & Co. Holding KG, Selb, Germany). Measurements were carried out in an argon atmosphere in the temperature range of 20–850 °C with a heating rate of 10 K·min^−1^ up to a temperature of 1000 °C.

Characterization of morphological properties (specific surface area, pore volume and pore size distribution) of the obtained activated carbons was performed by N_2_ adsorption–desorption at 77 K using a Quantachrome Autosorb Nova 2200e device and the NovaWin software package (Quantachrome Instruments, Boynton Beach, FL, USA). Degassing of the samples was carried out at a temperature of 170 °C. The Brunauer–Emmett–Teller (BET) specific surface area S_BET_ (m^2^/g) was determined from the adsorption isotherm in the range of relative nitrogen vapor pressures (0.05 ≤ p/p_0_ ≤ 0.3). The cumulative pore volume, V_total_ (cm^3^/g), was determined from the amount of nitrogen adsorbed at p/p_0_ ≈ 1 (p/p_0_ = 0.996). Parameters of the meso- and macro-porous structure of the synthesized carbons were calculated via the Barrett–Joyner–Halenda (BJH) method based on the modified Kelvin equation. This approach is considered valid for the capillary condensation theory and was used for the evaluation of the meso- and macro-pore size distribution (model of cylindrical pores) [41]. In parallel, the non-local density functional theory (NL–DFT) method of pore-size distribution calculation was used. This approach is based on the construction of adsorption isotherms in certain pore geometries using classical fluid density functional theory [42]. The pore size distribution was determined by solving the adsorption integral equation using regularization techniques. A slit pore model in NovaWin 10.0 software was applied in this paper. The surface morphology of the obtained activated carbons was determined using scanning electron microscopy (SEM, JSM–6700F, JEOL, Tokyo, Japan).

## 3. Results and Discussion

The thermogravimetric (TG) and derivative thermogravimetric (DTG) curves of raw coffee waste obtained in an argon atmosphere are shown in Figure 4. Fitting of the DTG curve with Gaussian functions (Figure 4) made it possible to determine the temperatures of the maximum intensity of chemical transformations, as well as the relative contents and temperature intervals of decomposition of the main components of raw plant materials.

Weight loss in the temperature range of 60–180 °C with a maximum at 100 °C is associated with the process of dehydration of the initial material without structural changes. The first stage of feedstock pyrolysis proceeds at temperatures between 200 and 380 °C with a maximum at 296 °C and is associated with thermal depolymerization of hemicellulose as the main sugar component of coffee waste, with an estimated content of 48.9%. The evaporation/decomposition of aromatic oils also occurs at this stage [43]. The decomposition of cellulose at a corresponding cellulose content of 17.4% occurs in the temperature range of 310 to 390 °C (stage 2), with the highest intensity at 345 °C, and is completely overlapped with the pyrolysis of hemicelluloses/oils. The final stage (stage 3) of thermally induced transformations is observed in a wide temperature range from 307 to 514 °C, reached at 403 °C and associated with the decomposition of lignin. The weight loss corresponding to the degradation of lignin makes it possible to evaluate the component of coffee grounds rich in aromatic rings (about 33.7%). However, above 520 °C and up to 1000 °C, only a slight weight loss (up to 6%) is observed, with a total of weight loss of about 78%. Nitrogen adsorption/desorption isotherms for S400 and S500 materials (Figure 5a,b) are type II isotherms according to the IUPAC classification [44], and both have hysteresis in the region of low relative pressures. The divergence of the adsorption and desorption branches at low pressures can be explained by the irreversible retention of nitrogen molecules in the carbon material powder, the size of which is close to that of the adsorbate molecules [45]. These isotherms are typical for meso- and macro-porous adsorbents and are determined by the physical sorption of the adsorbate on the outer surface of the particles. S400 carbon adsorption isotherms (Figure 5a) start from p/p_0_ ≈ 0, and the completion of the monolayer nitrogen coating occurs at p/p_0_ ≈ 0.05 and corresponds to a specific surface area (S_BET_) of about 31 m^2^/g (Table 1). For S500 carbon (Figure 5b), the inflection of the adsorption branch is relatively flatter, which indicates multilayer adsorption with incomplete formation of a nitrogen monolayer. The completion of the monolayer coating can be considered at the pressure range 0.12 < p/p_0_ < 0.19, which corresponds to S_BET_ ≈ 170 m^2^/g. A change in the shape of the isotherm with a further increase in relative pressures for both S400 and S500 samples indicates an increase in the thickness of the adsorbed nitrogen layer.

The surface of sample S400 is mostly uniform (Figure 5a,b). Large pores ranging in size from 0.1 to 1 µm are visually observed both on the surface and inside of carbon particles. The surface of the material is visually smooth, without corrosion caused by the thermochemical activation of potassium hydroxide, which is the reason for the relatively low specific surface area (Table 1).

The adsorption isotherms measured for samples S600 and S700 (Figure 5c,d) are similar and can also be assigned to type II according to IUPAC. The main difference between these samples is the volume of absorbed gas required for the formation of a nitrogen monolayer, which indicates an increase in the content of micropores for sample S700 (Table 1). At the same time, low-pressure hysteresis was observed for both samples. The evolution of the microporous structure for samples S800 and S900 causes high-pressure hysteresis of the H4 type (Figure 5e,f) according to the IUPAC classification [44], associated with capillary condensation of nitrogen in the micro- and meso-pores of the carbon material. The growth in the adsorption branch of isotherms for S800 and S900 carbons is due to multiple processes of evaporation and condensation of nitrogen at p/p_0_ ≈ 1.

The development of a porous structure is directly observed in the SEM images of the S900 sample (Figure 6c,d). Increasing the thermochemical treatment temperature leads to narrowing of the pores and the formation of root-shaped pores, resulting in a sponge-like carbon particle. Surface corrosion of carbon is caused by the thermochemical activation of potassium hydroxide and is the reason for the high specific surface area of the material (>1000 m^2^/g). It can be argued that an activation temperature in the range of 800–900 °C is optimal for obtaining a carbon material with a highly developed surface area based on the raw material composition (Table 1).

The EDS spectrum measured for the S900 sample indicates that the obtained material mainly contained carbon (about 93.46 at.%), oxygen (about 4.85 at.%) and potassium (about 1.31 at.%) elements. Respectively small quantities of silica (about 0.38 at.%) as well as the possibility of magnesium and calcium presence was detected.

A more detailed analysis of the development of meso- and micro-pores as a result of thermochemical activation was performed using complementary BJH and DFT methods [46]. Pore size distributions of the surface area and volume were calculated by modeling adsorption isotherms for 0.35 < p/p_0_ < 1 using the BJH approach (Figure 7). An increase in the thermochemical activation temperature from 400 to 500 °C causes an increase in the area by 50%, mainly due to a change in the area of mesopores. The main contribution to the pore surface area for S400 carbon is observed for pores with a diameter in the range of 3–4 nm. An increase in temperature to 500 °C causes structural changes with an increase in the content of pores of about 3.5 nm in size and a redistributive transition of 4 nm to 5 nm pores (Figure 7a). The next increase in the activation process temperature from 500 to 800 °C leads to a decrease in the area of pores with a size of 4 nm by 30%, as well as to an increase in the pore size from 5 nm to 6.5 nm (Figure 7a).

The observed evolution of the porous structure is due to the burnout of carbon particles along their contour surface, which leads to a decrease in pores with a size of 4 nm. Simultaneous burnout of the inner surface of the pores initiates an increase in the pore size up to 6.5 nm when the burnout of the pore walls causes the merging of several micropores into mesopores, which corresponds to a sharp increase in both the area (Figure 7a) and the volume of mesopores (Figure 7b). The observed changes in pore volume are correlated with the specific surface area. In the case of S400 and S700 carbons, pores of about 10 nm in size contribute to the total volume, the number of which decreases with a further increase in the activation temperature.

The specific surface area of mesopores is provided mainly by pores up to 7 nm in size, since at these sizes the S(d) curves reach a plateau (Figure 8a). In addition, an increase in the pore volume occurs in the entire studied range of sizes (Figure 8b), and no inflection is observed in the V(d) dependences. The change in the microporous structure of carbon samples as a result of thermochemical activation at various temperatures was studied using the DFT method for modeling nitrogen adsorption/desorption isotherms with the approximation of a slit-like pore shape.

The formation of micropores begins at 500 °C (Figure 9) when the material obtained at 400 °C has a mesoporous structure with only a surface area of 23 m^2^/g (Table 1, DFT method). It was noticed that in the samples obtained at temperatures of 500 to 800 and 900 °C there was an increase in the content of pores with a size of 0.65–1.25 nm, which provide up to 90% of the specific surface area and up to 80% of the total pore volume (Figure 9). In the case of a constant ratio of potassium hydroxide and raw materials in the production of carbon material, an important factor affecting the morphology and porous structure of the resulting sample is the temperature of thermochemical activation.

The surface roughness of the obtained carbons was characterized by the fractal dimension D by calculation based on nitrogen adsorption porosimetry data. For the case of a Euclidean surface, D = 2; however, for an irregularly developed surface, the D value can vary from 2 to 3 and thus expresses the degree of surface roughness and/or porous structure. To determine the fractal dimensions of a surface, several models based on the gas adsorption and desorption method can be used, including the Langmuir model, the Frenkel–Halsey–Hill (FHH) model and the thermodynamic model [47].

The fractal dimension of the obtained carbons was studied by the modified FHH method, which is used for multilayer adsorption [48]. The surface fractal dimension is determined by rearranging the nitrogen adsorption isotherms (Figure 5) and plotting the dependence of lgV on lg(lg(p_0_/p)) according to the following equation: lg(V/V_0_) = A(lg(lg(p_0_/p))) + const, where V is the volume of adsorbed gas, V_0_ is the volume of adsorbed gas corresponding to the formation of a monolayer of N_2_ molecules, p is the equilibrium pressure, p_0_ is the vapor pressure of saturated gas and A is the slope of the curve, which depends on the fractal dimension [47].

The dependences of lgV on lg(lg(p_0_/p)) were linearly approximated in different ranges of relative pressures (ranges 1 and 2 in Figure 10), corresponding to single-layer and multilayer adsorptions, respectively, and the values of slope A were determined for each sample. The values of fractal dimension D were calculated as D = A + 3 or D = 3A + 3. The van der Waals interactions of nitrogen molecules with the carbon surface dominate at low relative pressures (range 1, Figure 10). Surface tension at the liquid–gas interface at the initial stage of adsorption can be neglected. In this case, the relationship between parameters A and D is determined by the equation: D = 3A + 3. In the case of multilayer adsorption (range 2, Figure 10), the surface tension between gas and liquid prevails, and the equation transforms into D = A + 3 [49].

The calculated values of A and D parameters for both ranges are given in Table 2. It was determined that the calculated values of the fractal dimension for single-layer adsorption are in the range of 2.07 < D < 2.67. The maximum value D = 2.67 is observed for sample S800. The values of the fractal dimension obtained for multilayer adsorption are close to 3 for samples S800 and S900. The maximum value of D is also observed for S800. It can be concluded that the samples obtained at higher temperatures of thermochemical activation (800–900 °C) have a rough, disordered surface, with splitting of micropores to mesopores.

Pore narrowing can be expected at a certain distance from the surface, corresponding to the formation of a root-like porous structure. Carbon materials obtained at temperatures of 500–700 °C have a smoother surface, which corresponds to a fractal dimension close to 2 for single-layered sorption. The fractal dimension for sample S400 (D = 1.32) exceeds the limits of 2 < D < 3, which can be explained by the irreversible retention of nitrogen molecules on the surface of the carbon sample and the inexpediency of using the formula in the pressure range of monolayer formation. Therefore, the obtained D values for these materials in multilayer adsorption are in the range of 2.66–2.68. Carbons obtained at 600 and 700 °C have intermediate values of fractal dimension (2.07 and 2.25 for single-layer absorption and 2.83 and 2.89 for multilayer adsorption, respectively), which indicates the formation of a porous structure at these temperatures, combining 2D and 3D elements.

The novelty of the presented results in comparison with previously published data [50,51,52] lies in the use of a combined carbonization/activation synthesis route, as well as in a complex approach to the analysis of nitrogen adsorption porosimetry data and the establishment of general patterns of the effect of activation temperature on the fractal dimensions of activated carbons derived from coffee waste. In addition, the obtained results agree with the data in paper [8].

In further research, it is planned to introduce the developed technology for the production of porous carbon into production.

## 4. Conclusions

Activated carbon was prepared from waste coffee grounds by thermochemical activation with potassium hydroxide in the temperature range of 400–900 °C. These materials were characterized by adsorption/desorption of nitrogen, TG, DTG and SEM. Thermal analysis showed that the decomposition of hemicellulose, cellulose and lignin occurs in the temperature ranges of 200–380, 310–390 and 307–514 °C, with a relative content of the three sugars of 48.9, 17.4 and 33.7 mass%, respectively. For all the obtained samples, the presence of both micro- and meso-pores was observed, and the content of micropores increased with an increase in the thermochemical treatment temperature. The BET specific surface area simultaneously increased in the range of 400–1050 m^2^/g. Active carbons obtained at an activation temperature of 900 °C with a micropore content of about 95% have the maximum BET surface area. A comparative analysis of the evolution of the pore size distribution was carried out using the BJH and NL-DFT (slit-like pore approximation) approaches. Fractal dimension analysis based on the Frenkel–Halsey–Hill approach indicates a complex porous system formed in the carbon sample containing micro-, meso- and macro-pores. There is a nonlinear change in the fractal dimensions calculated for single-layer adsorption with a minimum (D = 2.02) for carbons obtained at 600 °C. At the same time, an increase in the activation temperature in the range of 400–900 °C causes an increase in the values of the fractal dimension calculated for single-layer adsorption from 2.68 to 2.97. The obtained results indicate that the formation of a developed porous structure during the thermochemical activation of coffee waste occurs in the temperature range of 700–800 °C. Tuning the structure and morphology of carbon materials is a promising way to optimize their electrochemical performance in aqueous electrolytes and improve the efficiency of charge storage systems.

## Figures and Tables

**Figure 1 materials-16-06127-f001:**
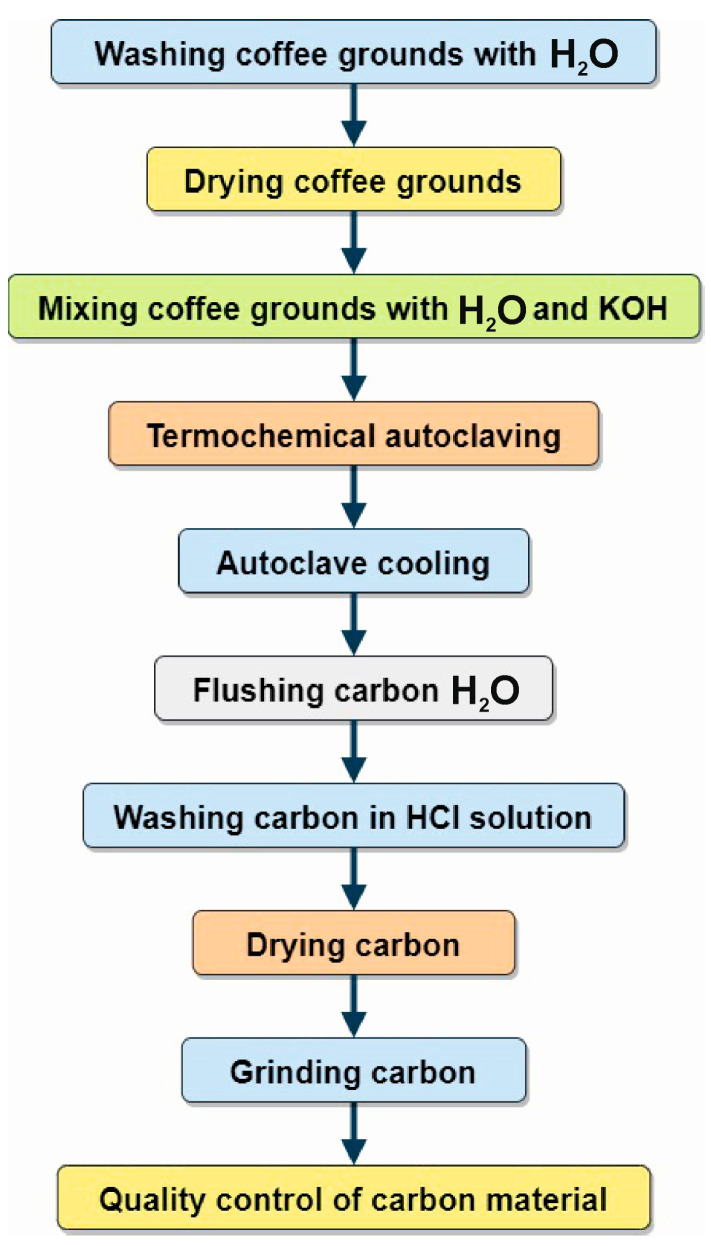
Technological scheme for the production of activated carbon from coffee grounds.

**Figure 2 materials-16-06127-f002:**
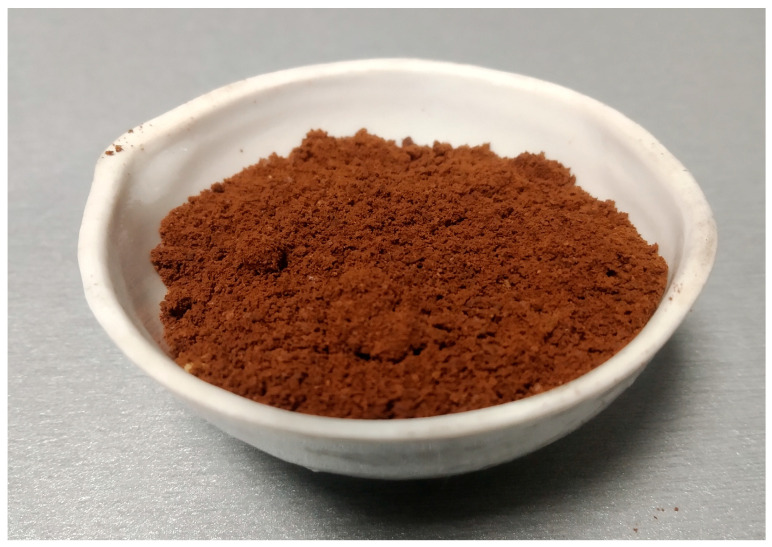
General view of the raw waste coffee grounds.

**Figure 3 materials-16-06127-f003:**
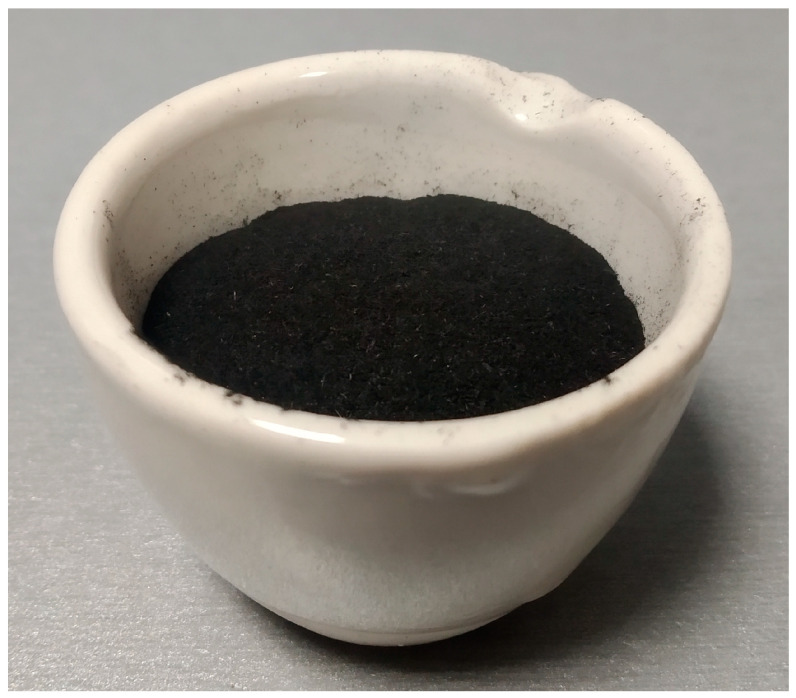
General view of the prepared activated carbon.

**Figure 4 materials-16-06127-f004:**
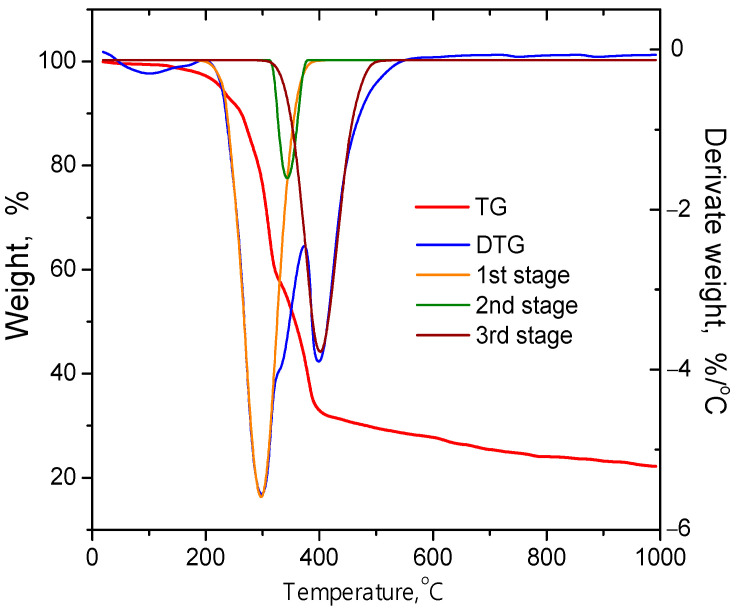
Mass loss (TG) and differential thermogravimetric analysis (TGA) curves obtained by heating waste coffee grounds at a rate of 10 °C/min.

**Figure 5 materials-16-06127-f005:**
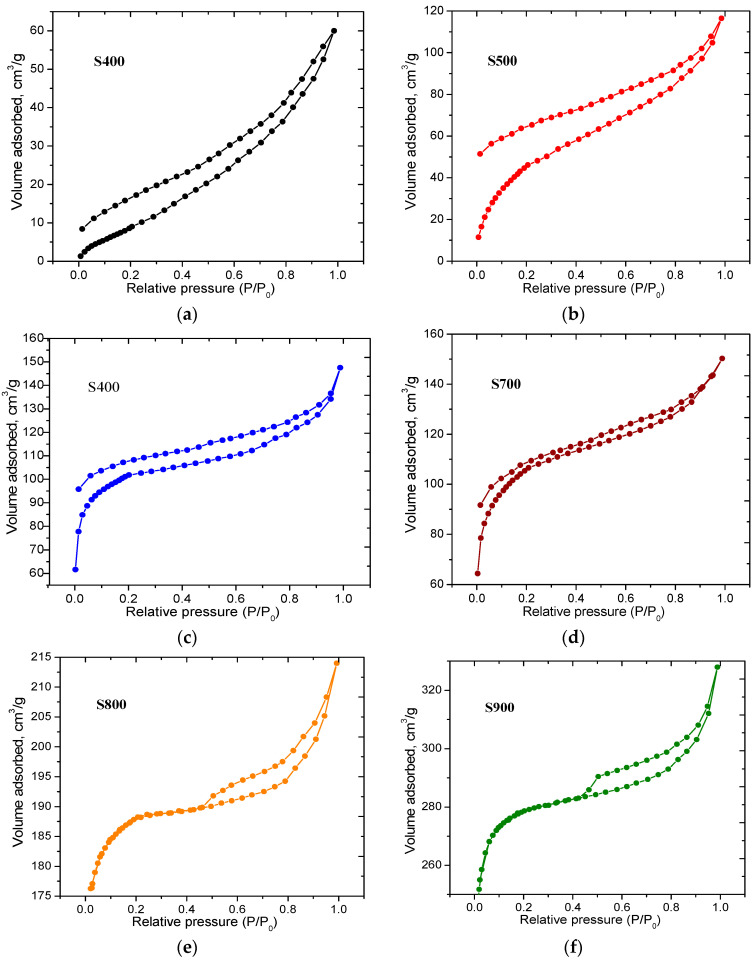
Nitrogen adsorption/desorption isotherms for carbons obtained at various temperatures: (**a**)–400 °C; (**b**)–500 °C; (**c**)–600 °C; (**d**)–700 °C; (**e**)–800 °C; (**f**)–900 °C.

**Figure 6 materials-16-06127-f006:**
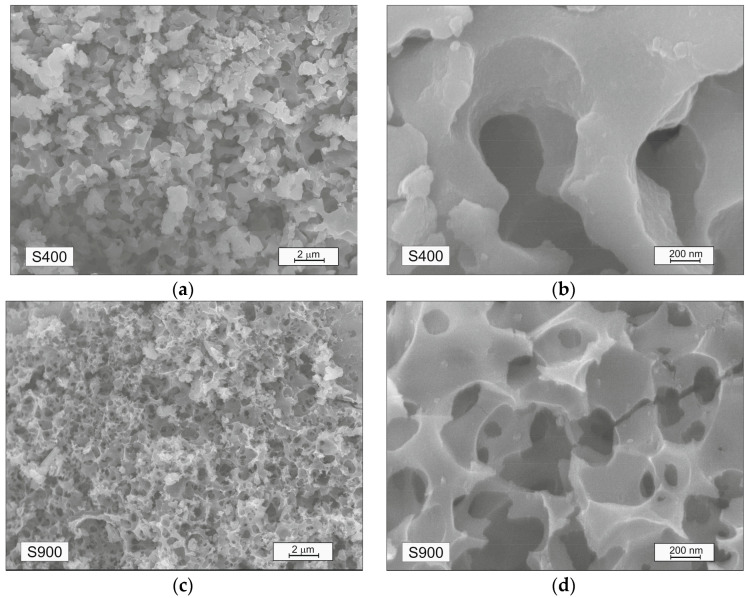
SEM images of (**a**,**b**) S400 and (**c**,**d**) S900 carbons at different magnifications and (**e**) EDS plot for S900 sample.

**Figure 7 materials-16-06127-f007:**
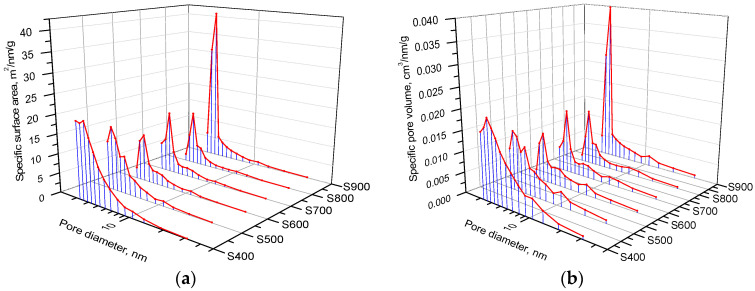
BJH distributions of (**a**) areas and (**b**) volumes of pores for carbon samples obtained at different temperatures.

**Figure 8 materials-16-06127-f008:**
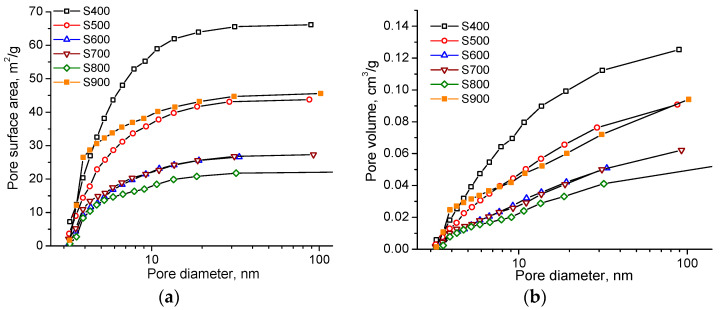
Dependence of (**a**) specific surface area and (**b**) volume of meso- and macro-pores by size for carbon samples obtained at different thermochemical treatment temperatures.

**Figure 9 materials-16-06127-f009:**
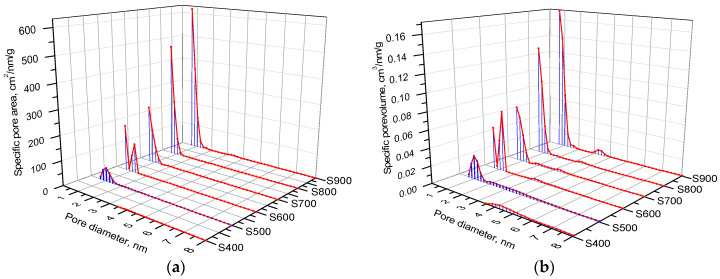
DFT distributions of pore (**a**) areas and (**b**) volumes for carbon samples obtained at different temperatures.

**Figure 10 materials-16-06127-f010:**
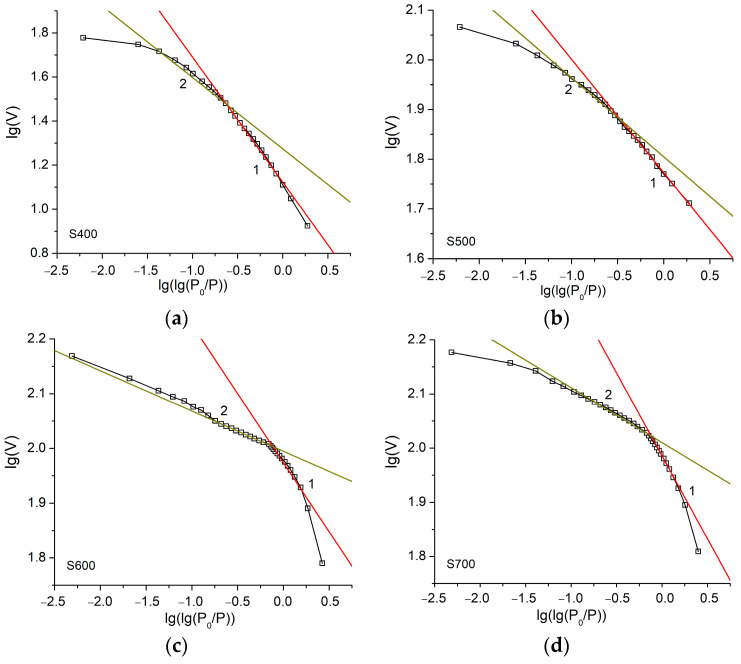
Dependences of lgV on lg(lg(p_0_/p)) for carbon samples obtained in different ranges of relative pressures (ranges 1 and 2) at different temperatures: (**a**)–400 °C; (**b**)–500 °C; (**c**)–600 °C; (**d**)–700 °C; (**e**)–800 °C; (**f**)–900 °C.

**Table 1 materials-16-06127-t001:** Morphological parameters of activated carbons obtained at different temperatures.

Sample	S400	S500	S600	S700	S800	S900
S_BET_, m^2^/g	31	172	374	446	703	1056
S_DFT_, m^2^/g	23	193	309	478	632	1170
S_meso_, m^2^/g	30	44	27	27	22	45
S_micro_, m^2^/g	–	80	319	402	671	996
V_total_, cm^3^/g	0.092	0.161	0.228	0.237	0.331	0.507
V_micro_, cm^3^/g	–	0.038	0.132	0.162	0.272	0.398

**Table 2 materials-16-06127-t002:** Values of fractal dimensions calculated for carbons obtained at different temperatures.

Sample	Range I Single-Layer Adsorption		Range II Multilayer Adsorption	
	A	D = 3A + 3	A	D = A + 3
S900	−0.19	2.43	−0.03	2.97
S800	−0.11	2.67	−0.02	2.98
S700	−0.31	2.07	−0.11	2.89
S600	−0.25	2.25	−0.17	2.83
S500	−0.23	2.31	−0.34	2.66
S400	−0.56	1.32	−0.32	2.68

## Data Availability

Data are contained within the article.

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
