# Peer review of "Porous Structure and Fractal Dimensions of Activated Carbon Prepared from Waste Coffee Grounds"

_materials, 2023, doi:10.3390/ma16186127_

Round 1
Reviewer 1 Report
General Comments: Sklepova et al. have worked on the manuscript entitled "Porous Structure and Fractal Dimensions of Activated Carbon Prepared from Waste Coffee Grounds". The authors have successfully synthesized porous carbons derived from coffee waste using a one-pot potassium hydroxide-assisted process, which is quite interesting. In particular, the use of DFT and BJH models for assessing the porosity and texture of materials is rarely discussed in the literature. The manuscript is well written, and all the studies were done competently. However, major revisions are required before this work can be accepted as material. Following revisions are necessary before final publication:
-
Similar reports published in the literature should be cited and discussed in the main text:
https://www.nature.com/articles/s41598-020-60625-y, Journal of CO2 Utilization 60 (2022), 101975, J. Colloid Interface Sci. 511 (2018), 259–267
2. Why did the author choose KOH as an activating agent over other agents published in the literature? Add a short description of it in the text.
- Elemental analysis should be included in the manuscript (subject to the availability of the resources).
- Supporting information should be provided along with the calculation methods, formulas used and short comparative table (including refence mentioned above) for the determination of the textural properties of these materials.
Editing is required.
Author Response
Dear Reviewer,
The answers are in the attached file.
Authors

Reviewer 2 Report
1- line 26, abstract section, porosimetry is usually called the technique for determining the distribution of pores but with mercury. The other is the adsorption of gases such as nitrogen to determine the specific surface area of materials. In this case they must correct the name mentioned in the abstract.
2- I consider it important to establish in the introduction the importance or justification of why the raw material for the production of carbon is the coffee residue.
3- line 115 page 3, They must write the units of measurement according to the international system of measurement. For example "h" instead of "hours". Review this error in the entire document for all the units of measure used and correct it according to this indication.
4- In figure 2, many curves are represented with the same color and if identified, I consider that this graph should be made clearer in order to understand it better. I suggest putting a legend for each curve or using different colors and guiding in the text what each one means.
5- adsorption-desorption isotherms are characteristic of both meso and macroporous materials, include this in the line 17 page 5.
6- Page 5: I believe that the surface area of the materials should be calculated by applying the BET equation, not in the way that the authors are reporting it. This needs to be fixed.
7- Tabla 1, there are numbers that should be supercript
8- Line 233 page 8, "adsorption" instead "absorption".
Author Response

(The authors gave the same response as above.)

Reviewer 3 Report
Manuscript materials-2567802 Round 1
Dr. Liubomyr Ropyak manuscript is devoted to results of a systematic study of the evolution of the morpho-23 logical properties of porous carbons derived from coffee waste using a one-pot potassium hydrox-24 ide-assisted process at a temperature in the range of 400–900°С. The manuscript is practical in nature, rich in experimental studies and I recommend it for publication with some remarks:
1. Keywords should not repeat words from the title and abstract. This decreases views and citation.
2. The use of abbreviations in the abstract, conclusion, and underdrawing captions is discouraged if possible.
Line 26: TG, DTG, SEM
Line 31: BJH and NL-DFT
3. Table 1: сm3/g authors forget upper index.
4. Figure 6: Why the origin of coordinates not the same for two graphs? It's hard to compare data.
5. Figure 1 It's not a bad idea to add a small visualization on the main technological operations.
6. Line 138: Which softwave authors use for NL–DFT method calculated pore size distribution?
7. Nowhere in the text full name of TG, DTG, BJH, NL-DFT and other not used.
8. Line 157: "...The evaporation/decomposition of aromatic oils also occurs at this stage..." Authors then start to describe the stages. But they don't mentioned 1st stage. In line 159 is just 2nd stage. There no any 1st stage in full text.
9. There is no uniform style in the design of charts, which complicates understanding and comparative analysis.
10. On Fig 4 it is not clear where the S400, and where the S900 is not a signature.
11. Line 216: "...The evolution of the microporous structure for samples S800 and S900 causes a high-pressure hysteresis of the H4 type (Fig. 4e,f)..." There is no Fig. 4e,f, only Fig. 4c,d. And what is mean H4 type? Nowhere in text authors did not disclose the concept
12. Line 343: "...The BET specific surface area..." what is BET?
13. It is not clear why this work is considered as an application for supercapacitors, as the potential advantages in morphology and surface development over classical activated carbon (1800cm2/g) are not shown, the most obvious application is as a method of utilization of coffee grounds for filtration and medical applications (utilization of coffee grounds is a serious environmental problem) in this context it is interesting to know whether the activation produces harmful substances or not.
14. I would recommend that authors put a lot of effort into the design of the paper. A lot of carelessness. For example, in Figure 1, the water formula is without a subscript.
15. Line 37-60. The first part of the introduction on energy resources is grossly overblown. The authors should reduce it to two sentences and concentrate on an overview of coffee-derived porous materials and their applications.
1. The work is full of pronouns. Please,rewrite sentences with pronouns. Personal pronouns aren't used in scientific works.
Author Response

(The authors gave the same response as above.)

Reviewer 4 Report
In the present paper entitled "Porous Structure and Fractal Dimensions of Activated Carbon Prepared from Waste Coffee Grounds", the authors report the evolution of the morphological properties of porous carbons derived from coffee waste. The following issues should be addressed to further improve the manuscript.
1. Experimental details are missing. The amounts and stoichiometry of materials and reagents for porous carbons derived from coffee waste have not defined thus the experiments are uncertain.
2. In the experimental section, the enzymatic/non-enzymatic degradation assessments of porous carbons need to be reported clearly.
3. Some data on storage stability (regarding their size, charge, and load stability) would add value to the study conducted.
4. The figures should be reorganized to fit the requirements of Materials. For example:
i) Please provide a scale bar in Figure 4.
ii) Please provide all captions with more comprehensive information.
5. The abbreviations should be further checked and provide the full title at the first appearance, and the writing and grammar need to improve.
6. In the manuscript, the following references may be considered:
DOI: 10.1039/D1NJ04123A
DOI: 10.1016/j.chphi.2023.100175
7. Discuss what are the limitations of this study?
8. Results have not been properly/sufficiently interpreted in the discussion. In general, it is better to make further comparisons between the measured parameters in this study and previous works done in similar studies in all parts.
-
Author Response

(The authors gave the same response as above.)

Round 2
Reviewer 4 Report
The authors addressed my concerns.